# evoKGsim+: a framework for tailoring Knowledge Graph-based similarity for supervised learning

Rita T. Sousa ✉, Sara Silva, Catia Pesquita

LASIGE, Faculdade de Ciências da Universidade de Lisboa, Portugal
{risousa,sgsilva,clpesquita}@ciencias.ulisboa.pt

**Abstract.** Knowledge graphs represent an unparalleled opportunity for machine learning, given their ability to provide meaningful context to the data through semantic representations. However, general-purpose knowledge graphs may describe entities from multiple perspectives, with some being irrelevant to the learning task. Despite the recent advances in semantic representations such as knowledge graph embeddings, existing methods are unsuited to tailoring semantic representations to a specific learning target that is not encoded in the knowledge graph.

We present evoKGsim+, a framework that can evolve similarity-based semantic representations for learning relations between knowledge graph entity pairs, which are not encoded in the graph. It employs genetic programming, where the evolutionary process is guided by a fitness function that measures the quality of relation prediction. The framework combines several taxonomic and embedding similarity measures and provides several baseline evaluation approaches that emulate domain expert feature selection and optimal parameter setting.

## 1 Introduction

Knowledge graphs (KGs) have been explored as providers of features and background knowledge in a wide variety of machine learning (ML) application scenarios [8]. One of these is predicting relations between KG entities that are not encoded in the KG, a problem cast as a classification task that takes as input a KG and a set of KG entity pairs. In the biomedical domain, ontologies are commonly employed to describe biological entities through semantic annotation. Tasks such as predicting protein-protein interactions using the Gene Ontology (GO) [12] or the mining of gene-disease associations on the Human Phenotype Ontology (HPO) [1] can be framed in this scenario.

In these cases, when we have a general-purpose KG (e.g., a KG that includes proteins and described their functions) that we aim to explore in the context of an independent and specific learning task (e.g., predicting if two proteins interact), it may very well be the case that large portions of the KG are irrelevant for the task. While in node/link/type prediction, instance representations such as embeddings [10] may be trained within the context of a particular learning task,

in our scenario, no such tuning is possible since the classification targets are not a part of the KG. This problem is exacerbated in complex domains, such as the biomedical, where KGs represent multiple views (or semantic aspects) over the underlying data, some of which may be less relevant to train the model towards a specific target. For instance, the prediction of protein-protein interactions using the GO is more accurate if only a portion of the ontology is used [9] (in this case, the one concerning biological processes).

This brings us to the challenge of tailoring the semantic representation (SR) of the KG entities to an independent and specific classification task when the classification target is not encoded in the KG. A KG-based SR is a set of features describing a KG entity obtained by processing the KG and bridge the gap between KGs and the typical vector-based representations of entities used by most ML techniques. Most state-of-the-art KG-based numeric representations are based on graph embeddings [10], which produce feature vector (propositional) representations of the KG entities. Taxonomic semantic similarity [4] can also be used as an SR by comparing entities based on the properties they share and their taxonomic relationships. Both types of approaches are, in fact, methods for feature generation, but they also result in feature selection by the heuristics and approaches they employ in creating the representations.

To address the specific goal of predicting relations between KG entities when those relations are not encoded in the KG, we postulate that similarity between the entities is a suitable frame for SR to be used by downstream supervised learning approaches. Then, the problem is how to tailor a given semantic similarity representation to the classification task, i.e., classifying a pair of entities as related or not. Previously, we presented evoKGsim [9], a methodology that learns suitable semantic similarity-based SRs of data objects extracted from KGs optimized for supervised learning. This tailoring is achieved by evolving a suitable combination of semantic aspects using Genetic Programming (GP) using taxonomy-based semantic similarity measures.

In this work, we present an extension of evoKGsim into a full framework, evoKGsim+, that encompasses 10 KG-based similarity measures based on a selection of representative state-of-the-art KG embeddings and taxonomic similarity approaches. We evaluate the framework in its full extension in benchmark datasets devoted to protein-protein interaction (PPI) prediction.

## 2   Methodology

evoKGsim+ targets classification tasks that take as input a KG and a set of KG individual pairs for which we wish to learn a relation that is outside the scope of the KG. The models are trained using external information about the classification targets for each pair. The evoKGsim+ framework is able to: (1) compute semantic similarity-based representations of KG individuals according to different semantic aspects and using different similarity approaches; (2) employ GP to learn a suitable representation targeted to a supervised learning task by combining the different semantic aspects; and (3) evaluate the outcome of

(2) against a set of static representations emulating experts. An overview of the framework is shown in Figure 1.

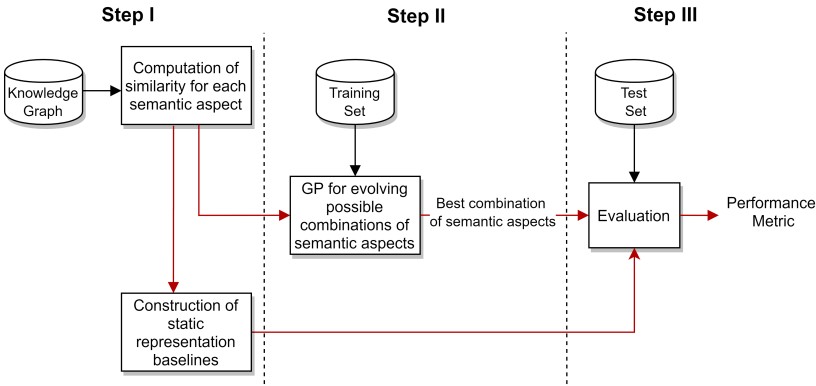

**Fig. 1.** Overview of the evoKGsim+ framework.

The first step of the framework is to represent each instance (i.e., a pair of KG entities) according to KG-based similarities computed for each semantic aspect. Currently, evoKGsim+ takes as semantic aspects the subgraphs rooted in the classes at a distance of one from the root class of the T-box in the KG, but this parameter can be easily adjusted. The second step is to employ GP to learn a suitable combination of the different aspect-based similarities, using a set of predefined operators, to address a given ML task. The last step is to evaluate the predictions made on the test set, and comparing them against optimized static representations that represent expert feature selection and parameter tuning.

This framework is independent of the specific implementation of KG-based similarity and the GP parameters employed to evolve the representations. Currently, evoKGsim+ supports 10 different KG-based similarity measures: 6 taxonomic similarity measures, derived by combining one of two information content approaches ($IC_{Seco}$ and $IC_{Resnik}$) with one of three set similarity measures (ResnikMax, ResnikBMA, and SimGIC [6]); 4 measures based on cosine similarity over embeddings generated from TransE[2], distMult[11], RDF2Vec [7] and Owl2Vec [5].

## 3  Evaluation

PPI prediction was chosen as our evaluation domain for the following reasons: (1) it is backed by a large ontology with multiple semantic aspects, the GO; (2) there are gold-standard datasets [3]; (3) it is well known that the GO aspects biological process (BP) and cellular component (CC) describe properties that are stronger indicators for PPI than the molecular function aspect (MF) [9], which provides an ideal test bed for the need of adapting the SR to the learning task.

Table 1 presents the results obtained using different similarity-based SRs. As baselines, we have used five static SRs (the BP, CC and MF single aspects, and the average and maximum of the single aspect similarities). The static SRs are based on a simple threshold-based classifier, where a similarity score for a protein pair above the threshold predicts a positive interaction. For evaluating the quality of a predicted classification, the weighted average F-measure (WAF) was used for stratified 10-fold cross-validation.

**Table 1.** Median of WAF for 10-fold cross-validation.

| Similarity Measure | Static SRs | | | | | evoKGsim |
|---|---|---|---|---|---|---|
| | BP | CC | MF | Avg | Max | |
| ResnikMax + $IC_{Seco}$ | 0.760 | 0.713 | 0.646 | 0.749 | 0.743 | **0.765** |
| ResnikMax + $IC_{Resnik}$ | 0.750 | 0.717 | 0.653 | 0.766 | 0.774 | **0.776** |
| ResnikBMA + $IC_{Seco}$ | 0.753 | 0.715 | 0.643 | 0.771 | 0.744 | **0.777** |
| ResnikBMA + $IC_{Resnik}$ | 0.753 | 0.714 | 0.648 | 0.777 | 0.772 | **0.782** |
| SimGIC + $IC_{Seco}$ | 0.736 | 0.682 | 0.642 | 0.729 | 0.701 | **0.746** |
| SimGIC + $IC_{Resnik}$ | 0.739 | 0.704 | 0.651 | 0.750 | 0.734 | **0.758** |
| TransE | 0.501 | **0.534** | 0.502 | 0.519 | 0.521 | 0.521 |
| distMult | 0.704 | 0.599 | 0.498 | 0.670 | 0.668 | **0.712** |
| RDF2Vec | 0.675 | 0.654 | 0.631 | 0.684 | 0.668 | **0.685** |
| Owl2vec | 0.678 | 0.662 | 0.621 | **0.693** | 0.686 | **0.693** |

evoKGsim with taxonomic similarity always achieves the best performance compared to the static SRs. Regarding the graph embedding approaches, TransE has performed worse than the other embedding methods. These differences are not unexpected since we are interested in learning which aspects of a KG are more relevant to the learning task, and most of the information to be processed is represented in the ontology portion of the KG, where taxonomic relations play an important role. Therefore, translational distance approaches that emphasize local neighbourhoods are less suitable than semantic matching methods, like disMult, or methods that capture longer-distance relations, such as path-based approaches (RDF2Vec and Owl2Vec).

When comparing the two SRs, evoKGsim with taxonomic similarity achieves a better performance than evoKGsim with embedding similarity. Although embeddings consider all types of relations, we hypothesize that taxonomic similarity can take into account class specificity that may give it the advantage over embedding similarity in more accurately estimating similarity.

## 4    Conclusion

We have developed a framework, evoKGsim+, that tailors KG-based similarity representations for supervised learning of relations between KG instances when the classification target is not encoded in the KG. We have shown that evoKGsim+ can generate tailored SRs that improve classification performance

over static SRs both using embedding similarity and taxonomic semantic similarity. This framework can be readily generalized to other applications and domains, where KG-based similarity is a suitable instance representation, such as prediction of drug-target interactions and gene-disease association, KG link prediction or recommendations.

**Acknowledgements** CP, SS, RTS are funded by the FCT through LASIGE Research Unit, ref. UIDB/00408/2020 and ref. UIDP/00408/2020. CP and RTS are funded by project SMILAX (ref. PTDC/EEI-ESS/4633/2014), SS by projects BINDER (ref. PTDC/CCI-INF/29168/2017) and PREDICT (ref. PTDC/CCI-CIF/29877/2017), and RTS by FCT PhD grant (ref. SFRH/BD/145377/2019). It was also partially supported by the KATY project which has received funding from the European Union's Horizon 2020 research and innovation programme under grant agreement No 101017453.

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
