# OpenReview forum: "evoKGsim+: a framework for tailoring Knowledge Graph-based similarity for supervised learning"
_eswc-conferences.org/ESWC/2021/Conference/Poster_and_Demo_Track — ESWC2021 P&D_

### Official Review · AnonReviewer1 · 2021-04-13
**Interesting, worthy of poster presentation**

**Rating:** 7
**Confidence:** 4

**Review:**

The authors have developed a framework called evoKGsim* that tailors KG-based similarity representations for supervised learning of relations between KG instances when the classification target is not encoded in the KG. evoKGsim* can generate tailored SRs that improve classification performance
over static SRs both using embedding similarity and taxonomic semantic similarity. The authors claim that the framework can be generalized.

Overall, I think the paper is interesting, though at times the writing is a little abstract and could use some examples. I think it deserves to be presented in the P&D track.

**Anonymity:**

Yes, I would like my review to remain anonymous.

---

### Official Review · AnonReviewer3 · 2021-04-14
**Promising demo**

**Rating:** 7
**Confidence:** 2

**Review:**

With the disclaimer that I am not an expert in KG similarity measures using ML techniques, the developed tool looks like a rather solid contribution to the research field.

The underlying idea of being able to capture the semantic information found in KG and transform that into a ML readable format strikes me as a very valuable contribution. The fact that it is a more advanced version of a tool already presented and documented, in combination with the strong performance in the evaluation makes me confident that the demo should be allowed to be presented at the conference.


**Anonymity:**

Yes, I would like my review to remain anonymous.

---

### Official Review · AnonReviewer4 · 2021-04-14
**An interesting but ML terminology-heavy contribution with promising results**

**Rating:** 7
**Confidence:** 2

**Review:**

This paper presents the framework evoKGsim* which implements several KG-based similarity measures, thus extending the initial evoKGsim methodology.

## Summary

Overall the paper is well-written.
However, the used terminology mostly from the ML domain makes it somewhat hard to follow, examples are listed in my detailed review.
Yet this can be resolved with some more explanations and rephrasing.

If some used terminology and the actual problem related to "learning targets" are clarified
this is a valuable and interesting contribution for this track.

## Introduction

"One of these [applications] is predicting specific relations between KG instances when the relation type is not encoded in the KG."

Should relations or relation types be predicted?
It's unclear to me if untyped relations exist for which types should be predicted, or if there are no relationships at all and relations should be predicted.

"since targets are not part of the KG"

It is unclear to me what a target is. This term was not introduced.
Is the target what is "supposed to be predicted"? Is the target an entity for which relations have to be predicted?

"This problem is exacerbated in complex domains, such as the biomedical, where KGs represent multiple views (or semantic aspects) ..."

Is this a domain-related problem? To me it sounds this is a problem/feature of RDF graphs in general.
Most RDF-based ontologies have this feature, i.e. the use of different vocabularies and terms to link resources.


(I assume that semantic representations are vector-based representations of KG entities)

The second paragraph starts with the following sentence:
"This brings us to the challenge of tailoring the semantic representation (SR) of the KG entities to an independent and specific learning task, for which the target is not encoded in the KG"

I'm not quite sure how to interpret this sentence, what is the target?
If semantic representations are some vector-based representations of entities in the KG, what is the tailoring then about? Which format do they need for learning tasks?
The tailoring is again mentioned later in this section but unfortunately did not clarify for me what it is about.

"Taxonomic semantic similarity kernels [3] can also be used as an SR ..."

What is a kernel in this context? There are several definitions, I couldn't find this term in the referenced work.

## Methodology

This section covers well what the framework is suppossed to do and what is implemented, additionally a figure is provided.


## Evaluation

The evaluation describes and justifies which prediction was chosen which is good.
Compared to the previous work more semantic similarity measures (SSMs) seem to be included, the contribution of this paper,
which are evaluated and achieve more performant results.

## Conclusion

Evaluations were performed on the Gene Ontology and generalizations are exemplified by drug-target interaction.
For the conclusion it might be interesting to mention generalizations beyond the biomedical domain with a little bit more details.

## Minor

* a small remark: there is currently a trend in the SemWeb field related to property graphs with names such as RDF* and SPARQL* (pronounced RDF star), thus, there might be confusions with the presented framework evoKGsim*, i.e. being a property-graph extension of evoKGsim

**Anonymity:**

Yes, I would like my review to remain anonymous.

---

### Official Review · AnonReviewer2 · 2021-04-15
**Need another paper to fully understand**

**Rating:** 6
**Confidence:** 4

**Review:**

This work is a rather interesting evolution of a previous one (evoKGsim, without *). Given the limited space, it is very difficult to fully understand it without recurring to the paper that introduced evoKGsim (Evolving knowledge graph similarity for supervised learning in complex biomedical domains). This is in my opinion the only issue with this paper, but it's also true that with the given space it was probably impossible to do otherwise. The 10 proposed measures that integrate the framework are rather known (at least for experts in the field), and I think that the discussion about the results obtained with these measures is quite interesting. It would be interesting to eventually add more measures to the framework, or to have a framework that allows to add measures as there are new additions quite often (TransE has its own successors such as TransH...)

**Anonymity:**

Yes, I would like my review to remain anonymous.

---

### Decision · Program_Chairs · 2021-04-19

Accept